# The Hippo Pathway in Metaplastic Breast Carcinoma: Prognostic Significance and Therapeutic Implications

**DOI:** 10.3390/cimb47121060

**Published:** 2025-12-18

**Authors:** Eleni Papamattheou, Alkistis Papatheodoridi, Ioannis Katsaros, Garyfalia Bletsa, Afroditi Nonni, Constantine Dimitrakakis, Dimitrios Haidopoulos, Angeliki Andrikopoulou, Areti Papakosta, Spyridon Marinopoulos, Aris Giannos, Sofia Koura, Eftychia Papachatzopoulou, Ioannis K. Papapanagiotou, Georgios I. Metaxas, Aikaterini-Gavriela Giannakaki, Meletios-Athanasios Dimopoulos, Flora Zagouri

**Affiliations:** 1First Department of Obstetrics & Gynecology, National and Kapodistrian University of Athens, “Alexandra” General Hospital of Athens, 11528 Athens, Greece; eleni.papamattheou@gmail.com (E.P.); dimitrac@ymail.com (C.D.); dimitrioshaidopoulos@gmail.com (D.H.); smarinopoulos@outlook.com (S.M.); arisgiannos@yahoo.gr (A.G.); sophia_koura@hotmail.com (S.K.); efipapaxatzopoulou@hotmail.com (E.P.); gpapamd@hotmail.com (I.K.P.); metaxakos@hotmail.com (G.I.M.); giannakaki.katerina@gmail.com (A.-G.G.); 2Department of Clinical Therapeutics, National and Kapodistrian University of Athens, “Alexandra” General Hospital of Athens, 11528 Athens, Greece; alkistispapath@gmail.com (A.P.); aggelikiandrikop@gmail.com (A.A.); papakostaar@gmail.com (A.P.); mdimop@med.uoa.gr (M.-A.D.); florazagouri@yahoo.co.uk (F.Z.); 3First Department of Surgery, National and Kapodistrian University of Athens, Laikon General Hospital, 11527 Athens, Greece; 4Hellenic Anticancer Institute, 11522 Athens, Greece; bletsag@yahoo.gr; 5First Department of Pathology, National and Kapodistrian University of Athens, 11527 Athens, Greece; afnonni@med.uoa.gr

**Keywords:** breast, breast cancer, Hippo pathway, metaplastic breast carcinoma, MpBC, targeted therapy, TAZ, YAP

## Abstract

**Background/objectives**: Metaplastic breast carcinoma (MpBC) is a rare, poorly differentiated breast cancer defined by the presence of ductal carcinoma along with areas of matrix-producing, spindle-cell, sarcomatous, or squamous differentiation. It does not express hormone receptors and has a poor overall prognosis. The Hippo molecular pathway was recently related to cancer progression and adjuvant therapy resistance. The objective of this study was to evaluate the expression of Hippo pathway transducers, YAP/TAZ, CCND1, and CTGF, in MpBC and their relation to the clinicopathological characteristics of the disease. **Methods**: Specimens from patients with MpBC treated at our department from 2003 to 2021 were analyzed utilizing immunohistochemistry and real-time PCR. **Results**: Forty-four female patients (62.6 ± 14.7 years old) met inclusion criteria and were included in this study. Strong nuclear YAP/TAZ expression was found in 61.4% of patients, while the expressions of CCND1 and CTGF were 3.9% and 12.5%, respectively. Patients presenting at an advanced stage had a statistically worse prognosis compared to the ones diagnosed with stage IA disease. Adjuvant chemotherapy was associated with better overall survival, while disease recurrence was significantly associated with a worse prognosis. **Conclusions**: Advanced stage at diagnosis and disease recurrence were significantly associated with worse prognosis in MpBC. However, adjuvant chemotherapy significantly led to better overall survival. The Hippo pathway is frequently deregulated (nuclear YAP/TAZ in 61.4% of patients), suggesting it is a compelling novel therapeutic target for this aggressive disease.

## 1. Introduction

Breast cancer is the most prevalent type of cancer among female patients, with approximately 300,000 new cases yearly in the United States of America [1]. Although much progress has been achieved in the treatment of invasive breast cancer, it remains the second cause of cancer-related death [2]. Metaplastic breast carcinoma (MpBC) is a rare type of breast cancer including a morphologically diverse group of breast cancer tumours and was officially recognized as a distinct pathology in 2000 [3].

MpBC constitutes approximately 0.2–5% of breast cancer cases and carries the worst prognosis regarding survival and local recurrence compared to the other breast cancer types [4]. It is composed of various cell types and, according to the World Health Organization, is classified into low-grade adenosquamous carcinoma, fibromatosis-like metaplastic carcinoma, squamous cell carcinoma, spindle cell carcinoma, metaplastic carcinoma with mesenchymal differentiation, myoepithelial carcinoma, and mixed metaplastic carcinoma [5]. The intertumoral heterogeneity of the disease could explain its aggressiveness and chemoresistance, and this neoplasm has a high tendency to present distant metastases to the brain, lungs and vertebrae [6].

Several signalling pathways are involved in the regulation of breast cells functions, with Hippo being one of the most prominent [7]. It is a highly complex network discovered in the 1990s in an experimental model of Drosophila melanogaster [8,9]. More specifically, this pathway plays an important role in organ size, tissue regeneration, wound healing, and stem cell self-maintenance by regulating cell proliferation and apoptosis [10]. The inactivation of this pathway leads to the transport of transcriptional coactivators, yes-associated protein (YAP) and transcriptional coactivator with PDZ-binding motif (TAZ), into the cell nucleus and their interaction with the target gene, resulting in carcinogenesis [9,11]. Furthermore, YAP/TAZ mediate the functional output of physical cues by activating gene transcription, particularly genes involved in matrix remodelling and cytoskeleton reorganization. Connective tissue growth factor (CTGF) and cyclin D1 (CCND1) are commonly induced by YAP/TAZ, and they encode matricellular proteins that serve as ligands for integrins and are functionally important for cell adhesion to matrices [12].

While the deregulation of the Hippo pathway has been extensively studied in other breast cancer subtypes, data regarding MpBC remain scarce. To date, limited research has investigated Hippo effectors in this specific neoplasm and the combined prognostic value of both YAP and TAZ, alongside their downstream targets, has yet to be fully elucidated in a larger population. The aim of this study was to evaluate the expression of Hippo pathway transducers, YAP/TAZ, CCND1 and CTGF, in female patients suffering from MpBC and elucidate their association with clinicopathological characteristics and the prognosis.

## 2. Materials and Methods

### 2.1. Study Protocol and Included Patients

This was carried out at “Alexandra” General Hospital, which is affiliated with the National and Kapodistrian University of Athens, Greece, and in accordance with a protocol agreed upon a priori by all the authors. The study protocol was approved by the Institutional Review Board (IRB) of Alexandra Hospital (ethical approval code 577; approval date: 20 January 2022). All participants signed an informed consent form, and the study was conducted following the principles of the Declaration of Helsinki [13].

All consecutive medical records of women treated at our department from 2003 to 2021 were retrospectively searched for patients diagnosed with MpBC. Eligibility was determined according to the following criteria: (i) adult patients (age >18 years old) treated at our department, (ii) completeness of data related to clinical features and treatment outcomes, (iii) sufficient biological materials for pathological and molecular analyses in diagnostic biopsies, and (iv) availability of pathology samples for a histology examination and PCR evaluation.

### 2.2. Study Variables

The following data were collected from all included cases: age at diagnosis, family cancer history, histopathologic evaluation (histological type, grade according to the Nottingham Grading System [14], TNM stage according to AJCC 8th edition [15]), immunohistochemical characteristics (estrogen receptor—ER, progesterone receptor—PR, HER2, Ki67 status), genetic testing for BRCA1 or BRCA2, treatment with surgery, chemotherapy in either neoadjuvant or adjuvant setting and adjuvant radiotherapy and data related to the follow-up of the patients (overall survival, disease free survival). Luminal A tumours were characterized by the presence of ER and/or PR and the absence of HER2, with a low expression of cell proliferation marker Ki-67 (<20%). Luminal B tumours were defined as positive for one of the hormone receptors with a high expression of Ki67 (greater than 20%), while HER-2 can also be positive. Triple-negative breast cancer (TNBC) is characterized as ER-negative, PR-negative, and HER2-negative [16]. Disease-free survival (DFS) was defined as the interval (in months) between the date of diagnosis and the date of recurrence or death or censored at last follow-up, whichever occurred first. Overall survival (OS) (in months) was defined as the interval between the date of diagnosis and the date of death or censored at last follow-up.

### 2.3. Immunohistochemistry

TAZ and YAP1 transducers expression in MpBC was detected on FFPE tissue using the rabbit anti-YAP and anti-TAZ antibodies, clone D24E4 of the ImmPress REAGENT KIT, UNIVERSAL, Anti-MOUSE/RABBIT Ig, VECTOR, (Cell Signaling Technology, Danvers, MA, USA) at a dilution of 1:50. Sections which were 4 μm thick, after deparaffinization in xylene and rehydration in graded ethanol solutions, were placed for 20 min in high-pH Tris-EDTA solution (DAKO, Copenhagen, Denmark) of heat-mediated antigen retrieval. Then, endogenous peroxidase blocking was performed by incubating the sections with H_2_O_2_, for 15 min at room temperature. After this step, sections were incubated firstly with the primary antibody overnight, and consequently with the secondary and tertiary antibodies. The visualization system used was the Immunologic Kit (BIOCARE, Pacheco, CA, USA). The chromogene used was DAB and the incubation time was 6 min. Finally, sections were counterstained with Mayer’s acid hematoxylin. The entire procedure was performed manually. As positive control, we used human lymphoma tissue, while negative control was assessed by omitting the primary antibody.

The topographic TAZ and YAP1 expression were evaluated by one pathologist (A.N.) in both the cytoplasm and the nucleus of the tumour cells, and in the main cellular components, namely normal breast, endothelial cells, non-lymphocytic stromal cells, and tumour-infiltrating lymphocytes (TILs), that were morphologically identified. In cancer cells, the evaluation of nuclear staining was based on the percentage of positive cells (0: negative, 1: 1–10% positive cells, 2: 10–50% positive cells and 3: >50% positive cells) and the staining intensity (low, moderate, strong). On this basis, TAZ/YAP positivity was defined as a distinct nuclear immunoreactivity in ≥10% of neoplastic cells [17]. Interestingly, nuclear expression was detected as well in some myoepithelial cells of adjacent normal breast tissue, while a low expression was seen in a few normal epithelial cells.

### 2.4. qRT-PCR

A skilled pathologist examined the hematoxylin and eosin (H&E)-stained slides to select blocks that had tumour tissue. The tumour portion of six shaved cuts (10 µm thickness) was subjected to RNA isolation using the PureLink™ FFPE RNA Isolation Kit (Thermofisher Scientific, Waltham, MA, USA), following the manufacturer’s protocol. Subsequently, total RNA (750 ng) was treated with DNase I (Invitrogen, Waltham, MA, USA). The RNA concentration was measured using a Qubit fluorometer 4.0 (Thermofisher Scientific, Waltham, MA, USA) and a Qubit RNA Broad range Assay kit (Thermofisher Scientific, Waltham, MA, USA) according to manufacturer’s instructions. The KAPA SYBR^®^ FAST One-Step qRT-PCR KIT Universal (KAPA Biosystems, Roche, Basel, Switzerland) was used for the relative quantification of CCND1 and CTGF mRNA expression, whereas ACTB was used as a housekeeping gene. One-Step qRT-PCR was performed in a final volume of 10 μL. Reaction mixture contained 1X KAPA SYBR^®^ FAST qPCR Master Mix, 200 nM of each forward and reverse primer (Table 1), 1X KAPA RT Mix, and 75 ng of total RNA template, while the cycling conditions were conducted according to manufacturer’s instructions. The reaction was performed using the LightCycler^®^ 480 Real-Time PCR system (Roche) [18,19].

### 2.5. Statistical Analysis

Statistical analysis was performed using the statistical package SPSS 26 (IBM company, Chicago, IL, USA, 2019). Parametric variables are presented by their mean values ± standard deviation (SD), whereas nominal ones are presented as frequencies and valid percentages. Normal distribution of collected data was based on the Central Limit Theorem and the size of the study’s sample and, thus, parametric tests were utilized. Student’s *t*-test was used for the comparison of two sample means, while ANOVA (Analysis of Variance) test was used for multiple comparisons. Pearson’s chi-squared test was utilized for the comparison of categorical variables. Kaplan–Meier curves were used for the estimation of the cumulative probability of death/recurrence and log-rank test was applied for their comparison. Hazard ratios (HRs) were determined using the Cox proportional hazards model and are reported with 95% confidence interval (CI). A *p*-value of <0.05 (two-tailed) was considered statistically significant.

## 3. Results

### 3.1. Patient Characteristics

Overall, 44 women with a mean age of 62.61 ± 14.74 years diagnosed with MpBC were enrolled in this study. Most patients included (56.8%) were postmenopausal. Two (4.5%) patients had a positive genetic test. Most of the participants reported a negative family cancer history (88.6%), whereas only five patients (11.4%) had a positive family history of lung, prostate, breast, or colorectal cancer among first-degree relatives. Detailed data regarding the demographic variables of the patients are demonstrated in Table 2.

### 3.2. Histology

In our results, mean tumour size was 3.7 ± 2.7 cm. The majority of patients were diagnosed at stage II (47.7% stage IIA, 22.7% stage IIB), while one patient (2.3%) presented with distant metastases (stage IV). Most tumours were triple-negative (86.4%), while both the estrogen receptor (ER) and progesterone receptor (PR) were positively expressed in four cases (9.1%). Human epidermal growth factor receptor 2 (HER2) was positively expressed in one case (2.3%). A high Ki-67 (>15%) was diagnosed at most patients (54.5%). Most tumours were grade 3 (86.4%). The most frequently diagnosed histopathological type was the mixed one (38.6%), followed by squamous cell carcinoma (22.7%), and metaplastic breast cancer with matrix-producing heterologous mesenchymal differentiation (20.4%).

### 3.3. Treatment Approach

All patients underwent a surgical resection of the primary tumour. Thirty-eight patients were treated with systemic adjuvant chemotherapy (86.4%), while twenty-nine patients also received adjuvant radiotherapy (65.9%).

### 3.4. YAP, TAZ, CCND1, CTGF Gene Expression

The nuclear YAP/TAZ staining was positive in 61.4% of the carcinomas analyzed (Figure 1), while the cytoplasmic expression of this pathway transducers was positive in 75.0% of tumours. Furthermore, the mean expression of CCND1 and CTGF genes were 3.9 ± 7.4% (median: 1%; 95% CI: 0.8–2.3%) and 12.5 ± 22.4% (median: 3.6%; 95% CI: 2.2–9.8%), respectively. No statistically significant association was observed between YAP/TAZ and CTGF-CCND1 expression and clinicopathological baseline characteristics, as shown in Table 3 and Table 4 and Figure 2.

### 3.5. Disease-Free Survival (DFS)

Regarding disease recurrence, local breast recurrence was observed in thirteen patients, while distant metastases developed in three patients. The median DFS rate was 60.00 (95% C.I. 36.56–83.44) months (Figure 3). Relevant hazard ratios of the examined variables are presented in Table 5.

### 3.6. Overall Survival (OS)

Over the follow-up period, 16 deaths were recorded. Median OS was not reached. Included patients had a mean OS of 149.24 (95% CI: 116.16–182.32) months (Figure 4). Patients with an advanced stage at diagnosis had a worse overall prognosis compared to women diagnosed with IA disease (regarding IIIB stage HR: 33.10; 95% CI 2.21–495.54; *p*-value: 0.01 and regarding IV stage HR:16.33; 95% CI: 1.31–203.61; *p*-value: 0.03). Adjuvant chemotherapy was a statistically significant protective factor (HR: 0.29; 95% CI: 0.09–0.91; *p*-value: 0.03). Finally, disease recurrence was significantly associated with worse prognosis (HR: 107.42; 95% CI: 1.65–6996.05; *p*-value: 0.03) (Figure 5). Detailed effects of examined variables on survival are shown in Table 6.

## 4. Discussion

Metaplastic Breast Cancer (MpBC) includes a highly diverse group of neoplasms characterized by an intimate admixture of adenocarcinoma with dominant areas of spindle cell, squamous, and/or mesenchymal differentiation. It is reported to have worse prognosis compared to the other types of breast carcinoma [21]. To our knowledge, this is the most conclusive study to investigate the association between the prognosis of MpBc patients and the deregulation of the Hippo pathway.

Demographic characteristics of women with metaplastic breast cancer vary, but most of studies agreed that it tends to occur more frequently in women over the age of 50 years old and it mostly concerns postmenopausal women [22,23,24,25]. Some cases of MpBC affecting male patients are also reported in the literature [23]. Our findings align with the existing literature, showing that MpBC typically affects postmenopausal women in their sixth decade. Furthermore, according to Pezzi et al., MpBC significantly affects African Americans and Hispanics compared to Caucasians [26].

MpBC tends to demonstrate unique histopathological and molecular features [4]. These tumours tend to be large in diameter (>5 cm), with negative axillary lymph nodes, and poor differentiation. Additionally, MpBC most commonly does not express hormone receptors (triple-negative phenotype) along with high Ki67 index and p53 positivity [27,28]. In our cohort, mean size was approximately 3.7 cm and about 9 out of 10 patients did not express any hormone receptor. The main histological types encountered were the squamous cell and mixed-type carcinomas, which are associated with a relative worse DFS and OS compared to invasive carcinoma NST [4,29]. Balkenhol et al. also showed that metaplastic carcinomas with mixed components and matrix-producing types have a higher risk of recurrence [30]. There are no specific radiological findings which characterize the MpBC, which usually presents with ill-defined borders on imaging studies, similar to other types of invasive breast cancer [31]. Although core needle biopsy is the gold standard method for the definite diagnosis of breast lesions, the diagnosis of this neoplasm can be quite challenging [32]. Many studies support that MpBC is susceptible to early hematogenous spread to distant organs such as the lungs, liver, and bones, while local recurrence is also quite frequent [33,34]. De novo metastatic disease was observed at >10% of the patients with MpBC, whereas the local or distant relapse concerned up to >50% of the cases [35]. Nevertheless, in our study only one patient primarily presented with stage IV disease (distant metastasis), thirteen patients experienced local recurrence, and sixteen developed metastases at a later course of their disease. Disease recurrence was a statistically significant factor affecting overall survival, but this result should be critically appraised due to the limiting number of included events.

Previously published reports have suggested that patients with MpBC have a poor response to chemotherapy and usually present with a large breast mass, and, thus, the most common surgical procedure for MpBC patients was modified radical mastectomy or mastectomy with or without axillary dissection [36]. On the other hand, Hu et al. analyzed 1665 patients with MpBC from the Surveillance, Epidemiology, and End Results database and concluded that those who underwent breast-conserving surgery had a better prognosis in early and locally advanced MpBC [37]. As far as systemic therapy is concerned, several studies underlined that the type of chemotherapy that was performed depended on the histological type of metaplastic breast cancer. Brown-Glaberman et al. reported that MpBC with sarcomatous elements responded to doxorubicin- and ifosfamide-based regimens [38].

It is very interesting to clarify the role of neoadjuvant chemotherapy in MpBC due to its resistance to chemotherapy, compared to other triple-negative tumours. Alongi et al. reported that only 20% of patients with MpBC responded to neoadjuvant chemotherapy compared to an 85% response rate of TNBC [39]. In our results, patients who received chemotherapy had significantly better survival. Unfortunately, the data related to the type of surgery and the choice of chemotherapy regimen were not available for this analysis. Regarding radiotherapy, the analysis of Hu et al. showed that it was associated with a better prognosis, but our results did not reveal any significant difference [40].

The Hippo pathway is crucial for regulating organ size, tissue homeostasis, and cell proliferation. YAP/TAZ are its major effectors. Incomplete phosphorylation allows YAP/TAZ to translocate into the nucleus, inducing cell proliferation and insensitivity to apoptosis signals, leading to tumour formation and metastasis [12,41]. EMT overexpression may also cause deregulation by affecting TAZ [42]. Downstream targets CCND1 and CTGF are often highly expressed in breast cancer and linked to poor prognosis in triple-negative cases [19,43]. The expression of TAZ/YAP has been reported, although to a different extent, in various BC subtypes including HR-positive BC, HER2-positive BC, and TNBC, and is related to resistance to chemotherapy and metastatic dissemination [10,44]. TAZ/YAP expression is found in various BC subtypes and is related to chemotherapy resistance and metastatic dissemination. Inactivation of tumour suppressor genes LATS1/2 or increased YAP/TAZ expression disrupts downstream targets, causing cancer cells to develop resistance against anti-cancer drugs [45]. Our study found that the Hippo pathway was deregulated in approximately 60% of the MpBC cases analyzed, suggesting that a majority of these patients could be candidates for therapies targeting this pathway. The high frequency of nuclear YAP/TAZ expression observed makes this pathway a compelling target. Diaz-Martin et al. assessed 15 patients with MpBC and, similarly to our results, concluded that 80% of them had a positive TAZ expression [46]. Given that MpBC is known for its chemoresistance, targeting the YAP/TAZ could represent a novel strategy to overcome this challenge [47]. The development of targeted therapies that disrupt the YAP/TAZ interaction is a promising area of research. Such therapies could potentially inhibit the oncogenic activity driven by the Hippo pathway. Furthermore, there is emerging evidence linking the Hippo pathway to the tumour microenvironment and immune evasion [48]. Therefore, targeting this pathway could not only have a direct anti-tumour effect but also potentially sensitize these tumours to immunotherapies, a key area of interest for triple-negative cancers like MpBC.

Women presenting with MpBC were associated with a relative worse survival compared to invasive ductal carcinoma [49]. Park et al. suggested that the poor prognosis of patients with MpBC is related to the development of hematogenous rather than lymphatic metastases [50]. Several researchers’ analyzing data from the Surveillance, Epidemiology, and End Results database compared the clinical characteristics and outcomes of patients with TNBC and MpBC and concluded that patients with MpBC had both worse disease-free survival and OS [26,49,51,52]. Major factors affecting prognosis were age at diagnosis, tumour size, histopathologic subtype, tumour grade, TNM stage, and hormone receptor status [30]. Our analysis showed that patients with an advanced stage at diagnosis had a worse overall prognosis compared to women diagnosed with IA disease. Even though most patients with Hippo pathway deregulation experienced relapse or died from the disease, no statistically significant correlations were found. Further long-term follow-ups may, however, be needed to draw more accurate conclusions about these clinical variables.

This study addresses a novel area by investigating the Hippo pathway in a well-defined cohort of MpBC patients, a topic with limited existing literature. However, as expected, our study is subject to certain limitations. Our data were retrospectively collected, and they rely on accurate record keeping. Additionally, the number of patients included was limiting and as a result a Type II statistical error could occur. Nevertheless, the number of patients included is in accordance with the prevalence of the disease. Collected data regarding treatment approach (e.g., type of surgery, sentinel lymph node status, neoadjuvant–adjuvant approach and regimens) were scarce, covered a long period of time, and thus, no safe conclusions could be reached regarding the optimal management of these tumours. Furthermore, specimens were examined by one pathologist and no interobserver validation was possible. No data were available regarding upstream regulators of the YAP/TAZ pathway and relevant mutations affecting it. Utilized antibodies assessed the total expression of YAP and TAZ, and an association of each individual molecule to MpBC was not possible. Despite these constraints, our study adds an important body of clinical data regarding MpBC and the significance of the Hippo pathway regulation in its progression. It provides an important foundation of clinical data on these rare breast neoplasms’ characteristics and outcomes. Future studies must prioritize the clinical translation of these molecular insights. This includes conducting prospective, multi-centre studies to analyze upstream molecular regulators of YAP/TAZ. Critically, this research should encompass clinical trials for targeted Hippo therapies and genetic-specific therapeutic agents, which are necessary to strengthen the available data and fully elucidate the factors affecting the clinical course and prognosis of MpBC. Furthermore, all clinicians should be aware of this rare clinical entity and be encouraged to report these cases to enhance the available evidence base and facilitate these crucial future investigations.

## 5. Conclusions

In conclusion, metaplastic breast cancer is a rare and aggressive subtype of breast cancer, accounting for only a small portion of all breast cancer cases. The study concluded that patient prognosis in MpBC is significantly worse with an advanced stage at diagnosis and the occurrence of disease recurrence. However, the use of adjuvant chemotherapy significantly led to better overall survival. A critical finding was the frequent deregulation of the Hippo pathway signalling (with strong nuclear YAP/TAZ expression in 61.4% of patients), which is generally associated with aggressive tumour behaviour, poor prognosis, and resistance to therapy in breast cancer. This deregulation suggests that the Hippo pathway is a compelling potential novel target for therapeutic intervention in this heterogeneous disease. Therefore, a better understanding of this pathway is crucial, and clinicians are encouraged to report these rare cases to enhance the evidence base for developing targeted therapies.

## Figures and Tables

**Figure 1 cimb-47-01060-f001:**
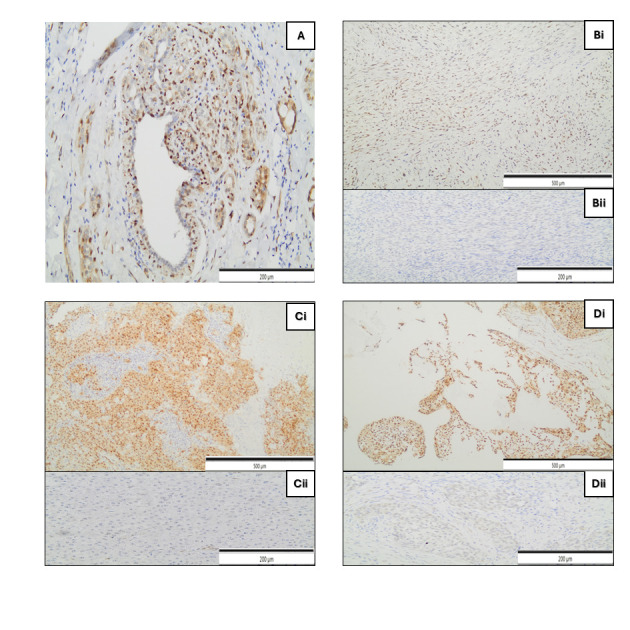
Immunohistochemical nuclear expression of TAZ/YAP in the following: (**A**) Normal breast tissue with positive myoepithelial and a few epithelial cells, ×200. (**B**) Spidle cell metaplastic breast carcinoma, with moderate to strong nuclear staining, ×100 (**Bi**); with no expression, ×200 (**Bii**). (**C**) Squamous cell metaplastic breast carcinoma, with mainly strong nuclear staining, ×100 (**Ci**); with no expression, ×200 (**Cii**). (**D**) Metaplastic breast carcinoma with heterologous mesenchymal differentiation showing mainly strong nuclear staining, ×100 (**Di**); with faint nuclear expression in a few neoplastic cells—negative, ×200 (**Dii**). Scale bars are included in the images for reference.

**Figure 2 cimb-47-01060-f002:**
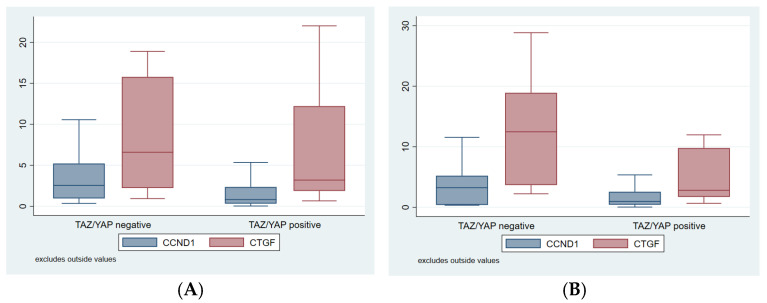
Box plots presenting CTGF and CCND1 expressions stratified by YAP/TAZ nucleus (**A**) and cytoplasm expression (**B**).

**Figure 3 cimb-47-01060-f003:**
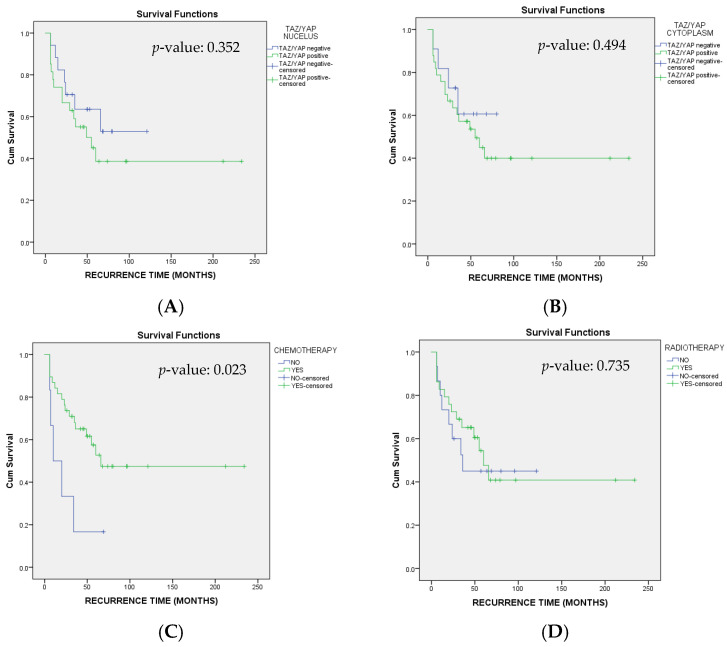
Kaplan–Meier survival curves regarding Disease-Free Survival (DFS) stratified by YAP/TAZ (**A**) nucleus and (**B**) cytoplasm expression, (**C**) chemotherapy and (**D**) radiotherapy. Log-rank test was applied for comparison and respective *p*-values are provided.

**Figure 4 cimb-47-01060-f004:**
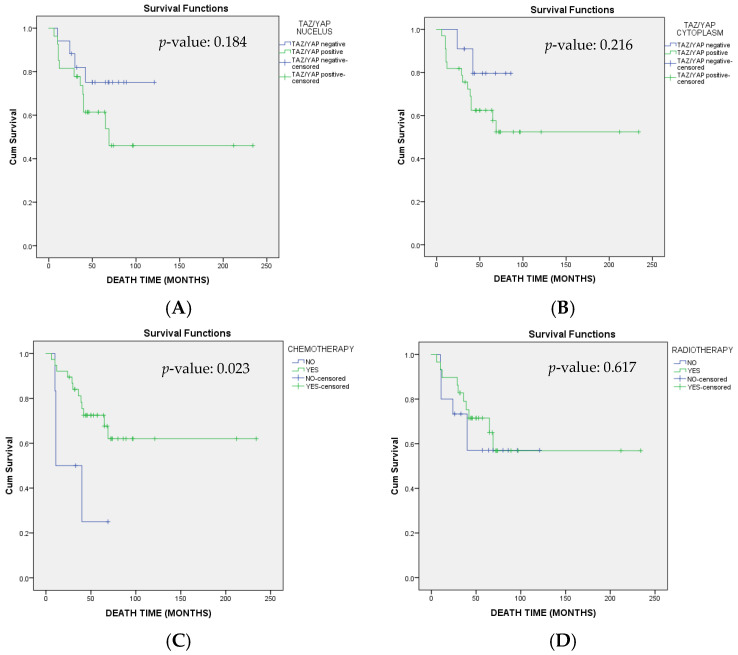
Kaplan–Meier survival curves regarding overall survival (OS) stratified by YAP/TAZ (**A**) nucleus and (**B**) cytoplasm expression, (**C**) chemotherapy and (**D**) radiotherapy. Log-rank test was applied for comparison and respective *p*-values are provided.

**Figure 5 cimb-47-01060-f005:**
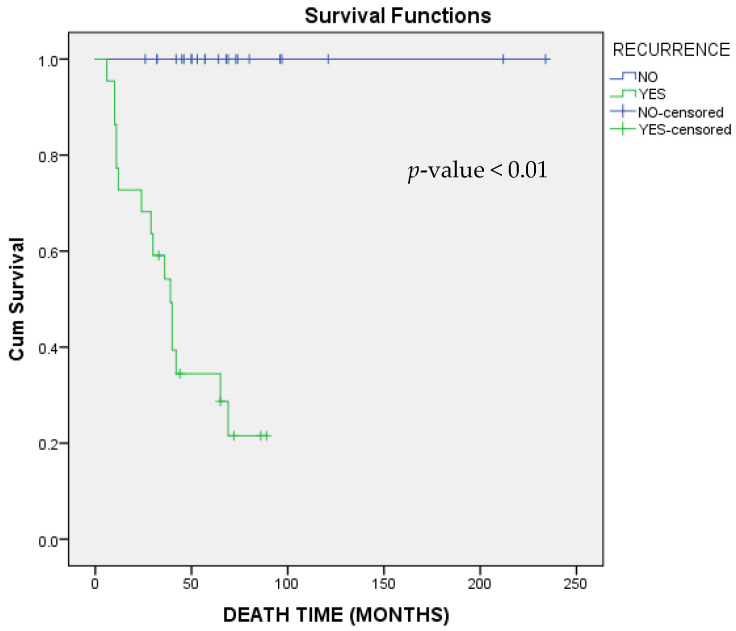
Kaplan–Meier survival curves regarding overall survival (OS) stratified by recurrence. Log-rank test was applied for comparison and respective *p*-value is provided.

**Table 1 cimb-47-01060-t001:** Real-time primer sequences.

Primers	Sequence (5′-3′)	Reference
ACTB F	TGGCACCACACCTTCTACAA	Sammarco et al., 2020 [19]
ACTB R	CCAGAGGCGTACAGGGATAG	
CCND1 F	ATCAAGTGTGACCCGGACTG	
CCND1 R	CTTGGGGTCCATGTTCTGCT	
CTGF F	CGAGCTAAATTCTGTGGAGT	Qi et al., 2005 [20]
CTGF R	CCATGTCTCCGTACATCTTC	

**Table 2 cimb-47-01060-t002:** Detailed included patients’ characteristics (n = 44).

Variable	N/Mean ± SD
Age	62.6 ± 14.7
Menopause Status	19 (43.2%)
Size (cm)	3.7 ± 2.7
TNM STAGE	
*Ia*	8 (18.2%)
*IIa*	21 (47.7%)
*IIb*	10 (22.7%)
*IIIa*	2 (4.5%)
*IIIb*	1 (2.3%)
*IIIc*	1 (2.3%)
*IV*	1 (2.3%)
ER (+)	4 (9.1%)
PR (+)	4 (9.1%)
Her2 (+)	1 (2.3%)
Ki67 (>15%)	24 (54.5%)
MOLECULAR SUBTYPE	
*TNBC*	38 (86.4%)
*Luminal A*	1 (2.3%)
*Luminal B*	4 (9.1%)
*Luminal B HER2*	1 (2.3%)
CHEMOTHERAPY	38 (86.4%)
RADIOTHERAPY	29 (65.9%)
GENETIC TEST	2 (4.5%)
FAMILY HISTORY	5 (11.4%)
HISTOLOGIC GRADE	
*GRADE 2*	6 (13.6%)
*GRADE 3*	38 (86.4%)
HISTOLOGIC SUBTYPE	
*Squamous n* (*%*)	10 (22.7%)
*Spindle cell n* (*%*)	8 (18.1%)
*Mixed*	17 (38.6%)
*MpBC with heterologous mesenchymal differentiation—Matrix-Producing n* (*%*)	9 (20.4%)
CCND1	3.9 (7.4%)
CTGF	12.5 (22.4%)
NUCLEAR YAP/TAZ (+) EXPRESSION	27 (61.4%)
CYTOPLASM YAP/TAZ (+) EXPRESSION	33 (75%)

**Table 3 cimb-47-01060-t003:** Positive nuclear and cytoplasmic YAP/TAZ expression stratified by histological and molecular subtype.

	YAP/TAZ (+) Expression			
	Cytoplasm	*p*-Value	Nucleus	*p*-Value
HISTOLOGICAL SUBTYPE				
*Squamous*	6 (60%)	0.588	7 (70.0%)	0.180
*Spindle*	7 (87.5)		6 (75.0%)	
*Mixed*	13 (76.5%)		7 (41.2%)	
*MpBC with heterologous mesenchymal differentiation—Matrix-Producing*	7 (77.8%)		7 (77.8%)	
MOLECULAR SUBTYPE				
*TNBC*	28 (73.7%)	0.195	23 (60.5%)	0.468
*Luminal A*	1 (100.0%)		1 (100.0%)	
*Luminal B*	4 (100.0%)		3 (75.0%)	
*Luminal B-like*	0 (0.0%)		0 (0.0%)	
TOTAL	33 (75.0%)		27 (61.4%)	

**Table 4 cimb-47-01060-t004:** CCND1 and CTGF expressions of included tumours.

	CCND1 (Mean ± SD)	*p*-Value	CTGF (Mean ± SD)	*p*-Value
HISTOLOGICAL SUBTYPE				
*Squamous*	4.67 ± 7.65	0.483	5.71 ± 2.29	0.853
*Spindle*	0.8 ± 0.83		7.28 ± 12.44	
*Mixed*	5.55 ± 9.9		15.61 ± 33.82	
*MpBC with heterologous mesenchymal differentiation—Matrix-Producing*	2.91 ± 3.62		13.44 ± 11.09	
MOLECULAR SUBTYPE				
*TNBC*	4.14 ± 7.8	0.683	13.87 ± 23.37	0.608
*Luminal A*	N/A		N/A	
*Luminal B*	1.43 ± 1.30		2.05 ± 0.97	
*Luminal B-like*	N/A		N/A	
TAZ/YAP NUCLEUS (+) EXPRESSION				
(*+*)	2.93 ± 5.64	0.253	8.89 ± 10.48	0.236
(*−*)	5.56 ± 9.49		19.53 ± 36.79	
TAZ/YAP CYTOPLASM EXPRESSION				
(*+*)	3.96 ± 8.27	0.987	11.98 ± 24.97	0.892
(*−*)	3.91 ± 3.94		12.85 ± 9.32	
(*−*)	5.56 ± 9.49		19.53 ± 36.79	

**Table 5 cimb-47-01060-t005:** Cox regression analysis for association of patients’ baseline characteristics with DFS.

Variable	HR	95.0% CI		*p*-Value
		Lower	Upper	
AGE	1.00	0.98	1.03	0.796
MENOPAUSE STATUS				
*No*	reference			
*Yes*	1.31	0.56	3.08	0.531
SIZE (cm)	1.04	0.88	1.22	0.652
TNM STAGE				
*Ia*	reference			
*IIa*	0.11	0.01	1.15	0.066
*IIb*	0.14	0.02	1.21	0.074
*IIIa*	0.14	0.01	1.37	0.091
*IIIb*	0.16	0.01	2.88	0.215
*IIIc*	1.83	0.11	31.78	0.678
*IV*	0.85	0.05	13.78	0.906
ER				
*Negative*	reference			
*Positive*	0.93	0.22	3.99	0.923
PR				
*Negative*	reference			
*Positive*	0.86	0.20	3.69	0.839
Her2				
*Negative*	reference			
*Positive*	0.05	0.00	944.48	0.544
Ki67				
*<15%*	Reference			
*>15%*	1.39	0.49	3.98	0.540
MOLECULAR SUBTYPE				
*TNBC*	reference			
*Luminal A*	5.64	0.69	46.03	0.107
*Luminal B*	0.98	0.23	4.22	0.973
*Luminal B-like*	0.00	0.00	-	0.983
CHEMOTHERAPY				
*No*	reference			
*Yes*	0.33	0.12	0.92	0.033
RADIOTHERAPY				
*No*	reference			
*Yes*	0.86	0.36	2.06	0.739
GENETIC TESTING (+)				
*Negative*	reference			
*Positive*	4.13	0.90	18.93	0.680
FAMILY HISTORY				
*Negative*	reference			
*Positive*	0.42	0.96	1.83	0.068
HISTOLOGIC GRADE				
*GRADE 2*	reference			
*GRADE 3*	1.53	0.52	4.53	0.443
HISTOLOGIC SUBTYPE				
*Squamous*	reference			
*Spindle cell*	1.10	0.24	4.89	0.908
*Mixed*	1.99	0.62	6.36	0.247
*MpBC with heterologous mesenchymal differentiation—Matrix-Producing*	1.99	0.52	7.57	0.313
TAZ/YAP NUCLEUS (+) EXPRESSION				
*Negative*	reference			
*Positive*	1.52	0.62	3.74	0.361
TAZ/YAP CYTOPLASM EXPRESSION				
*Negative*	reference			
*Positive*	1.45	0.49	4.30	0.502
CCND1	0.94	0.86	1.03	0.208
CTGF	1.00	0.99	1.02	0.797

**Table 6 cimb-47-01060-t006:** Cox regression analysis for association of patients’ baseline characteristics with overall survival.

Variable	HR	95.0% CI		*p*-Value
		Lower	Upper	
AGE	1.01	0.97	1.04	0.672
MENOPAUSE STATUS				
*No*	reference			
*Yes*	1.51	0.55	4.15	0.429
SIZE (CM)	1.10	0.92	1.32	0.298
TNM STAGE				
*Ia*	reference			
*IIa*	0.87	0.22	3.51	0.850
*IIb*	1.11	0.22	5.54	0.896
*IIIa*	2.09	0.22	20.23	0.524
*IIIb*	33.11	2.21	495.54	0.011
*IIIc*	9.61	0.86	108.04	0.067
*IV*	16.33	1.31	203.61	0.030
ER				
*Negative*	reference			
*Positive*	0.66	0.09	5.01	0.689
PR				
*Negative*	reference			
*Positive*	0.64	0.08	4.83	0.663
Her2				
*Negative*	reference			
*Positive*	0.05	0.00	22,666.23	0.647
Ki67				
*<15%*	reference			
*>15%*	0.97	0.28	3.34	0.967
MOLECULAR SUBTYPE				
*TNBC*	reference			
*Luminal A*	9.84	1.09	88.36	0.041
*Luminal B*	0.71	0.09	5.43	0.743
*Luminal B-like*	0.00	0.00	-	0.988
CHEMOTHERAPY				
*No*	reference			
*Yes*	0.29	0.09	0.91	0.033
RADIOTHERAPY				
*No*	reference			
*Yes*	0.77	0.28	2.13	0.620
GENETIC TESTING				
*Negative*	reference			
*Positive*	1.56	0.20	11.89	0.669
FAMILY HISTORY				
*Negative*	reference			
*Positive*	0.03	0.00	9.35	0.237
HISTOLOGIC GRADE				
*GRADE 2*	reference			
*GRADE 3*	0.43	0.14	1.33	0.142
HISTOLOGIC SUBTYPE				
*Squamous*	reference			
*Spindle cell*	1.56	0.22	11.10	0.657
*Mixed*	3.77	0.79	17.87	0.095
*MpBC with heterologous mesenchymal differentiation—Matrix-Producing*	3.98	0.70	22.59	0.119
TAZ/YAP NUCLEUS EXPRESSION				
*Negative*	reference			
*Positive*	2.11	0.68	6.56	0.196
TAZ/YAP CYTOPLASM EXPRESSION				
*Negative*	reference			
*Positive*	2.46	0.56	10.85	0.234
CCND1	0.93	0.82	1.06	0.272
CTGF	0.99	0.95	1.03	0.573
RECURENCE				
*No*	reference			
*Yes*	107.42	1.65	6996.04	0.03

## Data Availability

The original contributions presented in this study are included in the article. Further inquiries can be directed to the corresponding author.

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
