# Peer review of "The Hippo Pathway in Metaplastic Breast Carcinoma: Prognostic Significance and Therapeutic Implications"

_cimb, 2025, doi:10.3390/cimb47121060_

Round 1
Reviewer 1 Report
Comments and Suggestions for Authors
Overall Assessment
This study examines a significant yet underexplored topic: the function of the Hippo pathway in metaplastic breast carcinoma (MpBC), a rare and aggressive subtype of breast cancer. The study provides significant data correlating YAP/TAZ activation with the progression of MpBC and its potential therapeutic impact. The work is mostly well-organized, and the topic is important and relevant. However, prior to publication, the article requires further clarification and enhancement in certain aspects.
Main Comments
1. The study adds to the lack of research on Hippo pathway dysregulation in MpBC, but the introduction should explain how it is different from earlier studies. What is the important new finding, other than showing that YAP/TAZ is overexpressed?
2. The discussion should not just repeat what is already known; it should also discuss how the findings could help with future therapy strategies.
3. As time frame for the study is long (2003–2021) . Were there any changes to the diagnostic criteria or immunohistochemistry methods that could have affected the results?
4. The explanations of IHC methods are not clear enough. Please provide more details: Retrieval of antigens. Did you validate the antibodies? Were there positive and negative controls used to ensure the accuracy of the quality? Visualization method: specify the chromogen used for counterstaining and if automated or manual staining was utilized.
5. More clarifications are needed about how to score the YAP/TAZ expression. Please specify: How many pathologists scored the slides independently? Was the scoring blind? How were the differences resolved? Report the level of agreement between observers (for example, Cohen's kappa or the intraclass correlation coefficient). Explain how intensity and percentage were utilized to differentiate "positive" from "negative" cases—was an H-score ?
6. It is uncertain whether the data exhibited a normal distribution prior to conducting parametric tests. Please confirm whether normality was assessed (e.g., the Shapiro-Wilk test) and provide a rationale for employing parametric testing.
7. The way abbreviations are used is not consistent (for example, "DFS," "OS," "MpBC," etc.). Make sure that all abbreviations are spelled out the first time they are used and that they are used the same way throughout the text.
8. Scale bars should be in IHC images.
9. Showing typical examples of positive and negative staining next to each other, ideally for both cytoplasmic and nuclear localization, could also improve the presentation.
10. The reported HR (107.42) for disease recurrence is very high, which is probably because there aren't many events. Please confirm and consider giving a clearer explanation or re-analyzing.
The Kaplan-Meier survival curve (Figures 2 and 3) presents a singular survival function for the entire cohort. One of the primary objectives of the study is to ascertain the significance of Hippo pathway markers (YAP/TAZ) in predicting outcomes. To achieve this, the authors should include stratified survival curves that show the differences between groups with high and low expression. These comparisons would help us understand better if marker expression is related to overall survival and the time between recurrences. For more clarity, please provide the log-rank p-values and number-at-risk tables.
12. Although YAP/TAZ nuclear expression was common, the absence of statistical association with clinicopathological characteristics should be investigated further. What could elucidate the absence of significant associations?
13. The Cox regression table is mostly clear, but the names of the "reference" categories should be the same. Please include a notation stating that "reference" refers to the baseline group for HR calculation.
14. The paper is mostly clear, but it needs to be fixed for grammar and style mistakes.
15. A lot of the discussion paragraphs on Hippo signaling and EMT discuss the same points; please avoid redundancy.
16. The conclusion should be more specific. It must emphasize the prospective advantages of Hippo-targeted therapy while also making it clear that this is still a hypothesis based on the data we have.
17. Please verify the references to ensure that they all have the same format.
Author Response
Overall Assessment
This study examines a significant yet underexplored topic: the function of the Hippo pathway in metaplastic breast carcinoma (MpBC), a rare and aggressive subtype of breast cancer. The study provides significant data correlating YAP/TAZ activation with the progression of MpBC and its potential therapeutic impact. The work is mostly well-organized, and the topic is important and relevant. However, prior to publication, the article requires further clarification and enhancement in certain aspects.
Comment 1:
Main Comments
- The study adds to the lack of research on Hippo pathway dysregulation in MpBC, but the introduction should explain how it is different from earlier studies. What is the important new finding, other than showing that YAP/TAZ is overexpressed?
Response 1: Thank you for your valuable comment. We revised conclusions (Abstract and main manuscript) section to better reflect the main findings of our study.
Comment 2: 2. The discussion should not just repeat what is already known; it should also discuss how the findings could help with future therapy strategies.
Response 2: Thank you for your input. We revised our discussion and further perspectives sections.
Comment 3: 3. As time frame for the study is long (2003–2021) . Were there any changes to the diagnostic criteria or immunohistochemistry methods that could have affected the results?
Response 3: Although the time frame of the study is quite long, the diagnostic criteria for the histological subtypes, that are included in it [spindle cell, squamous, with heterologous mesenchymal differentiation and mixed (metaplastic and conventional adenocarcinomatous elements)] haven’t been seriously changed. Concerning the immunohistochemistry method, it has been automated by the time, and the antibodies panel for the diagnosis may have been expanded. However, the main diagnostic immunohistochemical markers are the same. Thus, we consider that there is no affection in the study’s results.
Comment 4:4. The explanations of IHC methods are not clear enough. Please provide more details: Retrieval of antigens. Did you validate the antibodies? Were there positive and negative controls used to ensure the accuracy of the quality? Visualization method: specify the chromogen used for counterstaining and if automated or manual staining was utilized.
Response 4: Thank you for your comment. We added relevant information to our Methods section to make IHC clearer and more reproducible.
Comment 5: 5. More clarifications are needed about how to score the YAP/TAZ expression. Please specify: How many pathologists scored the slides independently? Was the scoring blind? How were the differences resolved? Report the level of agreement between observers (for example, Cohen's kappa or the intraclass correlation coefficient). Explain how intensity and percentage were utilized to differentiate "positive" from "negative" cases—was an H-score ?
Response 5: Thank you for your input. An expert pathologist assessed the YAP/TAZ expression. We revised the Methods section to clarify your concerns.
Comment 6: 6. It is uncertain whether the data exhibited a normal distribution prior to conducting parametric tests. Please confirm whether normality was assessed (e.g., the Shapiro-Wilk test) and provide a rationale for employing parametric testing.
Response 6: Thank you for your comment. The number of patients included (44) was sufficient to apply the Central Limit Theorem and conduct parametric tests. No specific test was carried out to check the normality of our data. We added a relevant section to Methods.
Comment 7:7. The way abbreviations are used is not consistent (for example, "DFS," "OS," "MpBC," etc.). Make sure that all abbreviations are spelled out the first time they are used and that they are used the same way throughout the text.
Response 7: Thank you for your comment. We reviewed the manuscript and made sure all abbreviations are used consistently and are explained at the first time mentioned.
Comment 8: 8. Scale bars should be in IHC images.
Response 8: Thank you for your comment. We revised IHC images and added scale bars for reference.
Comment 99. Showing typical examples of positive and negative staining next to each other, ideally for both cytoplasmic and nuclear localization, could also improve the presentation.
Response 9: Thank you for your input. We added representative IHC figures in Figure1.
Comment 10:10. The reported HR (107.42) for disease recurrence is very high, which is probably because there aren't many events. Please confirm and consider giving a clearer explanation or re-analyzing.
Response 10: Thank you for your comment. Indeed, the limited number of patients with recurrence in combination with the small number of total deaths in follow-up period lead to this result. Nevertheless, it is clinically expected to see such a result. We further analyzed it in our discussion and included Figure 5 to better visualize it.
Comment 11: The Kaplan-Meier survival curve (Figures 2 and 3) presents a singular survival function for the entire cohort. One of the primary objectives of the study is to ascertain the significance of Hippo pathway markers (YAP/TAZ) in predicting outcomes. To achieve this, the authors should include stratified survival curves that show the differences between groups with high and low expression. These comparisons would help us understand better if marker expression is related to overall survival and the time between recurrences. For more clarity, please provide the log-rank p-values and number-at-risk tables.
Response 11: Thank you for your comment. We added Kaplan-Meier curves stratified by YAP/TAZ nucleus and cytoplasm expression in respective Figures.
Comment 12: 12. Although YAP/TAZ nuclear expression was common, the absence of statistical association with clinicopathological characteristics should be investigated further. What could elucidate the absence of significant associations?
Response 12: Thank you for your input. We revised the discussion section to better address this issue. The extremely rare nature of this disease was a major limitation and could potentially lead us to false negative results, as we acknowledge in the limitations section. Nevertheless, we included all consecutive patients with metaplastic breast cancer treated at our department from 2003 to 2021 and this limited sample size is consistent with the overall rarity of the disease, even when considering the over 10,000 breast cancer patients treated at our hospital during the same period.
Comment 13:
- The Cox regression table is mostly clear, but the names of the "reference" categories should be the same. Please include a notation stating that "reference" refers to the baseline group for HR calculation.
Response 13: Thank you for your comment. We revised both Tables to clearly state reference categories.
Comment 14: 14. The paper is mostly clear, but it needs to be fixed for grammar and style mistakes.
Response 14: Thank you for your comment. A native speaker reviewed the manuscript and corrected grammar and syntax mistakes.
Comment 15: 15. A lot of the discussion paragraphs on Hippo signaling and EMT discuss the same points; please avoid redundancy.
Response 15: Thank you for your input. We reviewed the whole Discussion section and omitted any redundancies.
Comment 16: 16. The conclusion should be more specific. It must emphasize the prospective advantages of Hippo-targeted therapy while also making it clear that this is still a hypothesis based on the data we have.
Response 16: Thank you for your comment. We critically revised conclusions section.
Comment 17: 17. Please verify the references to ensure that they all have the same format.
Response 17: Thank you for your comment. We used EndNote reference manager to apply journal’s format to References.
Reviewer 2 Report
Comments and Suggestions for Authors
The topic of this paper is interesting and clinically relevant, as the deregulation of the Hippo pathway in metaplastic breast carcinoma (MpBC) is still poorly characterized. The manuscript provides valuable preliminary data; however, several aspects should be improved to enhance the scientific rigor and impact of the study.
- the sample size (n = 44) is relatively small, which limits the statistical power and the generalizability of the conclusions. The authors are encouraged to consider expanding the dataset, possibly by including additional cases from collaborating institutions or by validating the findings using publicly available datasets such as TCGA, METABRIC, or SEER (also through kaplan meyer plotter).
- the survival analysis could be further developed. Kaplan–Meier survival curves should be provided, stratified by key clinicopathological and molecular variables (e.g., nuclear YAP/TAZ expression, CCND1 and CTGF levels, treatment modalities such as chemotherapy and radiotherapy). The inclusion of Kaplan–Meier plots with log-rank p-values and hazard ratios (HR with 95% CI) would substantially improve the clarity and robustness of the prognostic associations.
- in Figure 1, the representative immunohistochemical images show YAP/TAZ expression not only in tumour tissue but also in normal breast epithelium and myoepithelial cells. This raises methodological questions about how expression levels were quantified and normalized. The authors should clarify whether tumoural YAP/TAZ expression was normalized to normal breast tissue and specify the method used (e.g., relative intensity, percentage of positive cells). If no normalization was performed, it is possible that YAP/TAZ activity presents similar expression levels in normal and tumour breast tissues, which could affect the interpretation of the results. It would also be useful to include additional representative images for each histological subtype and for positive vs. negative staining patterns in the Supplementary Materials to better illustrate inter-tumour variability.
- the real-time PCR data should be shown in more detail, ideally through bar plots or box plots displaying relative CCND1 and CTGF expression levels, together with the corresponding statistical comparisons. Presenting these results would enhance transparency and allow readers to better assess the relationship between molecular and immunohistochemical findings.
Overall, the study addresses an important topic and provides an interesting foundation, but these additions would greatly improve the scientific value and clarity of the manuscript.
Comments on the Quality of English LanguageThe overall quality of the English language is acceptable and the manuscript is generally understandable. However, several sentences would benefit from clearer structure and more concise phrasing.
Author Response
The topic of this paper is interesting and clinically relevant, as the deregulation of the Hippo pathway in metaplastic breast carcinoma (MpBC) is still poorly characterized. The manuscript provides valuable preliminary data; however, several aspects should be improved to enhance the scientific rigor and impact of the study.
Comment 1:
- the sample size (n = 44) is relatively small, which limits the statistical power and the generalizability of the conclusions. The authors are encouraged to consider expanding the dataset, possibly by including additional cases from collaborating institutions or by validating the findings using publicly available datasets such as TCGA, METABRIC, or SEER (also through kaplan meyer plotter).
Response 1: Thank you for your comment. Indeed, the number of included patients is limiting to lead to statistically significant results. However, we included all consecutive patients with metaplastic breast cancer treated at our department from 2003 to 2021 and this limited sample size is consistent with the overall rarity of the disease, even when considering the over 10,000 breast cancer patients treated at our hospital during the same period. This study is an original study based on our patients’ data and integrating these results with available databases would alter the purpose of our study. Further systematic reviews and meta-analysis could explore this topic.
Comment 2: the survival analysis could be further developed. Kaplan–Meier survival curves should be provided, stratified by key clinicopathological and molecular variables (e.g., nuclear YAP/TAZ expression, CCND1 and CTGF levels, treatment modalities such as chemotherapy and radiotherapy). The inclusion of Kaplan–Meier plots with log-rank p-values and hazard ratios (HR with 95% CI) would substantially improve the clarity and robustness of the prognostic associations.
Response 2: Thank you for your input. We added Kaplan-Meier figures regarding OS and DFS stratified by these categories.
Comment 3: in Figure 1, the representative immunohistochemical images show YAP/TAZ expression not only in tumour tissue but also in normal breast epithelium and myoepithelial cells. This raises methodological questions about how expression levels were quantified and normalized. The authors should clarify whether tumoural YAP/TAZ expression was normalized to normal breast tissue and specify the method used (e.g., relative intensity, percentage of positive cells). If no normalization was performed, it is possible that YAP/TAZ activity presents similar expression levels in normal and tumour breast tissues, which could affect the interpretation of the results. It would also be useful to include additional representative images for each histological subtype and for positive vs. negative staining patterns in the Supplementary Materials to better illustrate inter-tumour variability.
Response 3: Thank you for your valuable comment. The nuclear staining in the normal breast tissue was identified mainly in some myoepithelial cells, a finding not referred in the literature, as far as we know, and in a low percentage of normal epithelial cells, with a low intensity. Because of this finding and for normalization purposes, the cut-off of the positive cases was defined in ≥10% of neoplastic cells, with moderate to strong staining intensity.
It’s important to mention, that the majority of the positive cases were shown a high percentage (>50% of positive cells), with a strong staining intensity. We included extra IHC figures in Figure 1.
Comment 4: the real-time PCR data should be shown in more detail, ideally through bar plots or box plots displaying relative CCND1 and CTGF expression levels, together with the corresponding statistical comparisons. Presenting these results would enhance transparency and allow readers to better assess the relationship between molecular and immunohistochemical findings.
Response 4: Thank you for your feedback. Box plots regarding CCND1 and CTGF expression are added.
Comment 5:
Overall, the study addresses an important topic and provides an interesting foundation, but these additions would greatly improve the scientific value and clarity of the manuscript.
Response 5: Thank you for you feedback that significantly improved our study.
Round 2
Reviewer 1 Report
Comments and Suggestions for Authors
We appreciate you submitting a revised version of your study and addressing numerous previous suggestions. The manuscript is easier to read, the figures are better, and the extra methodological information makes it easier to reproduce the work. The stratified survival curves and the amended commentary make the study scientifically valuable.
However, there are still some comments that need to be made before the study may be accepted:
1. The Study's Novelty (Comment 1)
Your response shows that you changed the conclusions; however, the issue is especially with the introduction.
Please explicitly say: • What existing research on YAP/TAZ in MpBC has found, and • How your work is different in terms of sample size, technique, or scientific contribution.
This is necessary for placing the work in the context of the current literature.
2. IHC Scoring Methodology (Comment 5)
The new changes say that one professional pathologist did the scoring. But further explanation is needed: • Was the scoring done without knowing the clinical outcomes? • Why was there only one scorer? • Was it impossible to figure out interobserver reliability (like kappa)? • If you can't conduct any more scoring, please make it clear that this is a methodological restriction.
3. Normality and the Application of Parametric Tests (Comment 6)
You mention the Central Limit Theorem, but it would make the approaches stronger if you could quickly say whether visual inspection or other diagnostic tests were employed to confirm that parametric assumptions were followed.
4. Very High Hazard Ratio (Comment 10)
Even if you have provided an explanation, it would be beneficial to include a clear statement in the discussion: • The large confidence interval indicates statistical instability, and • This result should be interpreted cautiously due to the limited number of events.
Author Response
Comment 1: We appreciate you submitting a revised version of your study and addressing numerous previous suggestions. The manuscript is easier to read, the figures are better, and the extra methodological information makes it easier to reproduce the work. The stratified survival curves and the amended commentary make the study scientifically valuable.
However, there are still some comments that need to be made before the study may be accepted:
- The Study's Novelty (Comment 1)
Your response shows that you changed the conclusions; however, the issue is especially with the introduction.
Please explicitly say: • What existing research on YAP/TAZ in MpBC has found, and • How your work is different in terms of sample size, technique, or scientific contribution.
This is necessary for placing the work in the context of the current literature.
Response 1: Thank you for your input that significantly improved our manuscript. We included a statement highlighting these aspects of our study in Introduction.
Comment 2: IHC Scoring Methodology (Comment 5)
The new changes say that one professional pathologist did the scoring. But further explanation is needed: • Was the scoring done without knowing the clinical outcomes? • Why was there only one scorer? • Was it impossible to figure out interobserver reliability (like kappa)? • If you can't conduct any more scoring, please make it clear that this is a methodological restriction.
Response 2: Thank you for your comment. An expert pathologist, Professor Nonni, independently assessed the specimens, without knowing the clinical outcomes. Thus, no interobserver difference is applicable. We added this as a limitation of our study.
Comment 3: Normality and the Application of Parametric Tests (Comment 6)
You mention the Central Limit Theorem, but it would make the approaches stronger if you could quickly say whether visual inspection or other diagnostic tests were employed to confirm that parametric assumptions were followed.
Response 3: Thank you for your feedback. No other specific test or visualization method was carried out to check the normality of data. We revised the relevant section in Methods to better address this issue.
Comment 4: 4. Very High Hazard Ratio (Comment 10)
Even if you have provided an explanation, it would be beneficial to include a clear statement in the discussion: • The large confidence interval indicates statistical instability, and • This result should be interpreted cautiously due to the limited number of events.
Response 4: Thank you for comment. We added a relevant section in Discussion highlighting this limitation.
Reviewer 2 Report
Comments and Suggestions for Authors
The authors follow the suggestions and improve some unclear point.
Now it can be accepted in present form.
Comments on the Quality of English LanguageThe overall quality of the English language is acceptable and the manuscript is generally understandable. However, several sentences would benefit from clearer structure and more concise phrasing.
Author Response
Comment 1: The authors follow the suggestions and improve some unclear point.
Now it can be accepted in present form.
Response 1: Thank you for your valuable feedback that significantly improved our manuscript.